# Combinatorial CRISPR screen identifies fitness effects of gene paralogues

Nicola A. Thompson[1], Marco Ranzani[1,5], Louise van der Weyden [1,5], Vivek Iyer[1], Victoria Offord[1], Alastair Droop[1], Fiona Behan [1], Emanuel Gonçalves [1], Anneliese Speak [1], Francesco Iorio [1,2], James Hewinson[1], Victoria Harle[1], Holly Robertson[1], Elizabeth Anderson[1], Beiyuan Fu[1], Fengtang Yang [1], Guido Zagnoli-Vieira [3], Phil Chapman[4], Martin Del Castillo Velasco-Herrera[1], Mathew J. Garnett [1], Stephen P. Jackson [3] & David J. Adams [1✉]

Genetic redundancy has evolved as a way for human cells to survive the loss of genes that are single copy and essential in other organisms, but also allows tumours to survive despite having highly rearranged genomes. In this study we CRISPR screen 1191 gene pairs, including paralogues and known and predicted synthetic lethal interactions to identify 105 gene combinations whose co-disruption results in a loss of cellular fitness. 27 pairs influence fitness across multiple cell lines including the paralogues *FAM50A/FAM50B*, two genes of unknown function. Silencing of *FAM50B* occurs across a range of tumour types and in this context disruption of *FAM50A* reduces cellular fitness whilst promoting micronucleus formation and extensive perturbation of transcriptional programmes. Our studies reveal the fitness effects of *FAM50A/FAM50B* in cancer cells.

[1] Wellcome Sanger Institute, Wellcome Trust Genome Campus, Cambridge, UK. [2] Human Technopole, Milano, Italy. [3] Wellcome Trust/Cancer Research UK Gurdon Institute, Cambridge, UK. [4] Cancer Research UK, Manchester Institute, Manchester, UK. [5]These authors contributed equally: Marco Ranzani, Louise van der Weyden. ✉email: da1@sanger.ac.uk

A major precept of cancer genetics is that normal cells acquire somatic mutations that provide a fitness advantage and drive tumour evolution and growth. Since these alterations are generally not found in normal cells, they may be exploited therapeutically, either by direct pathway inhibition, for example, in the context of activated oncogenes, or via synthetic lethality. The concept of synthetic lethality was pioneered by the yeast and *Drosophila* genetics communities who realised that the disruption of multiple genes simultaneously could elicit cell death in situations where disruption of such genes singly did not[1]. In more recent times, synthetic lethality has been exploited as an approach to treat cancers, the most notable example being the development of PARP inhibitors to treat patients with *BRCA1/2* mutant tumours[2–4]. Although substantial efforts have been made to identify synthetic lethal (SL) interactions in human cancer cells, we are still a significant way from having a genome-wide map of cancer dependencies, with many established SL interactions appearing to be highly context dependent[5]. Thus, systematic screens for SL gene pairs represent a powerful approach to define interactions that may be exploited in the clinic and to understand the context dependencies in which they operate.

The mammalian genome has evolved to carry parallel pathways or in some cases multiple redundant genes on which cells rely for survival—the principal reason for these gene sets seems to be to buffer and protect cells from the adverse consequences of gene loss, allowing cells to operate with higher fidelity and/or plasticity, even under stressful conditions. One set of these genes are the paralogues; genes derived from a common ancestral gene that now reside in different regions of the genome[6–9]. As above, several theories have been proposed for the creation of paralogues, one being that paralogues have developed to create functional redundancy, presumably as a result of selective pressure. Examples of essential paralogues include *RPL22L1* and *RPL22*, which are ribosomal proteins, and *YAP1/WWTR1(TAZ)* in the Hippo pathway[9]. Importantly, although several paralogue dependencies have been established, we are currently unable to accurately predict which paralogue pairs may be essential or functionally related by their sequence alone. Intriguingly, many essential paralogue pairs are part of the same protein complex[8]. A good example of this are components of the BAF/PBAF complexes, such as *ARID1A/ARID1B* and *SMARCA2/SMARCA4*, which are required for a range of processes such as transcription and chromatin regulation[10,11]. Disruption of these gene pairs results in growth arrest and cell death; phenotypes which appear to be influenced by cell lineage and differentiation status[12].

As noted above, cancer cell line screens are powerful tools for the identification of SL interactions because they can be used to systematically and comprehensively screen the genome without making any prior assumptions about which genes interact[5,13]. Generally, these screens have been performed using either compounds/targeted-agents, shRNA/siRNAs, or more recently with CRISPR[14–17]. Many of these screens have been performed in the context of specific genetic changes, such as in panels of cells with defined genetic alterations, or in isogenic cell lines, so that genetic interactions can be readily identified[15,18]. More recently, it has become possible to use paired gRNAs, also known as combinatorial or multiplex CRISPR screening, to identify essential gene pairs[19–24]. This approach to exploring genetic epistasis facilitates the identification of gene combinations that are SL by screening at scale. Here, we deployed CRISPR screening to interrogate 1191 gene pairs including 645 paralogues, 447 mutually exclusive genetic interactions defined using mutual exclusivity modelling of cancer data, and a set of 95 literature curated SL pairs. Our screening of two melanoma cell lines (A375 and MeWo) and a retinal epithelial line (RPE-1) identified 27 SL pairs occurring in ≥2 cell lines. This included the poorly characterised Family with sequence similarity 50 Member A & B (*FAM50A/FAM50B*) gene pair, whose disruption precipitates a loss of cellular fitness associated with apoptosis and widespread dysregulation of transcription. *FAM50A/FAM50B* are particularly notable among our collection of genetic interactions because ~4% of cancers profiled by the TCGA show loss of *FAM50B* expression (0–10% across tumour categories), thus highlighting the *FAM50A/FAM50B* axis as a potential therapeutic target.

## Results

**Selection of gene pairs for combinatorial screening.** Gene pair sets were chosen to be included in our library based on three distinct biological rationales (Fig. 1A). The first of these was a set of putative SL partners derived from two published bioinformatic analyses of human mutation and expression data[25,26], where pairs of genes had been identified as 'co-lost' less frequently than expected by chance. We intersected the gene sets from these studies, resulting in 447 overlapping candidate pairs for which gRNAs could be designed. The second gene set consisted of the 95 highest scoring gene pair interactions for which gRNAs could be designed as defined by a curated database of SL interactions (SynLethDB)[27]. Notably, this set of genes includes pairs derived from a vast array of biological contexts and tumour types, allowing us to assess if these interactions were essential in our system and in the cell lines we screened. Our library also included gRNAs for four gene pairs to test interactions between *PARG1/XRCC1*, *KDM6A/UTY*, *KDM6A/KDM6B* and *KDM6B/UTY*, implicated from genome sequencing studies[28]. The final gene set consisted of paralogous pairs. To define these genes, we built a computational pipeline to identify paralogue pairs (two-member paralogue families) with >20% DNA sequence homology/similarity and filtered this collection to identify genes where there was a single common orthologue in either *Caenorhabditis elegans* (Wormbase; WS251) or *Drosophila melanogaster* (Flymine; FB2015_05) and where disruption of this gene resulted in death of the organism. In this way, we identified 701 gene pairs, 645 of which were amenable to targeting by CRISPR (see Methods). In order to assess the performance of our library we also included a panel of established essential and non-essential genes[29,30]. A complete list of all gene pairs is provided in Supplementary Data 1.

**Library design, construction and cell line screening.** Our library was constructed using a dual promoter system (human U6 and synthetic U6) and was assembled using Gibson cloning[31]. We designed 3–5 guides for each gene and placed guides together as pairs and also paired each of them with a non-targeting/control guide, to allow assessment of both guide–guide and single-guide activity (Fig. 1B; Supplementary Data 11–4 and Methods). Prior to library construction, the efficiency of the paired gRNA construct was confirmed by fluorescence-activated cell sorting (FACS), utilising guides against two cell surface markers (CD15 & CD33)[32] (Supplementary Fig. 1 and Supplementary Data 2). We used our library to screen two deeply characterised melanoma cell lines (A375 and MeWo), and RPE-1 cells that are near-diploid and non-transformed and thus represents a 'normal' comparator. After lentiviral integration of Cas9, Cas9 activity (≥90%) of each cell line was confirmed using a reporter assay[32] (Supplementary Fig. 2). Screens were performed in technical triplicate at 1000× representation for a total of 28 days, harvesting cells for DNA extraction and sequencing at day 14 and 28 (see Supplementary Fig. 3 and Methods). Baseline values for gRNA abundance were generated by infecting a non-Cas9-positive cell line in triplicate in matched conditions to the dropout screen. These cells were harvested at day 7.

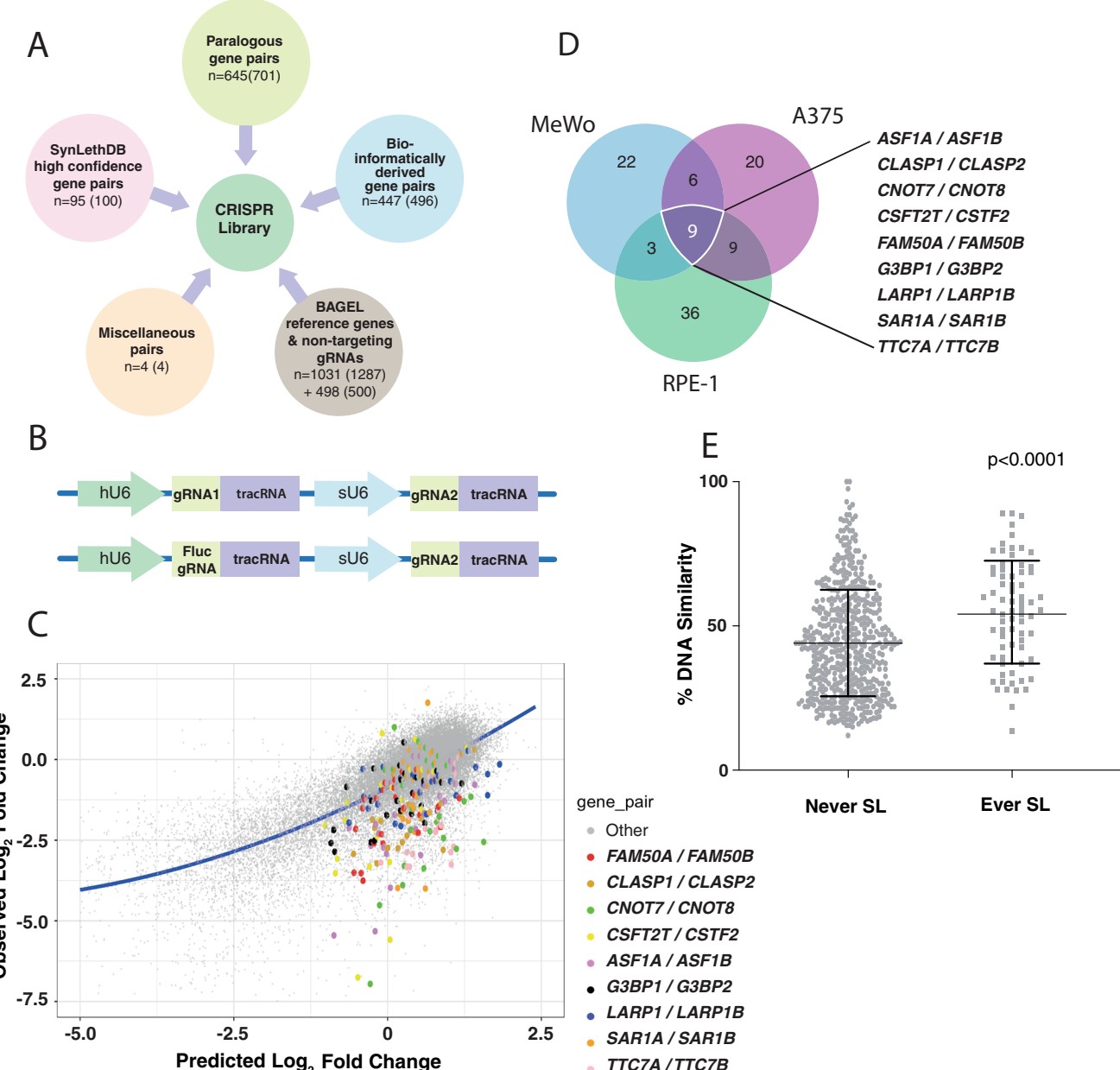

**Fig. 1 Combinatorial CRISPR screen for cancer targets. A** Schematic of the genes and gene pairs included in the paired gRNA library. The number of gene pairs for which the minimum guide number could be identified with our gRNA selection criteria is shown. The number in brackets is the total number of pairs in each category. **B** Design of the multiplex gRNA construct. To assess single-guide activity, a non-targeting gRNA (*Fluc*) was placed under the hU6 promoter and the gRNA against the candidate gene was placed under the sU6 promoter. To assess paired gene activity, all five gRNAs for one gene were under the hU6 promoter, whereas the guides targeting the other gene were under the sU6 promoter. **C** Derivation of residual values. The predicted log₂-fold change (FC) of the paired construct was calculated from the sum of the growth phenotypes of individual guides and the observed (experimentally determined) log₂FC was the actual behaviour of the pair in the screen. The behaviour of the population was modelled using Loess smoothing (blue line). The residual was calculated as the vertical difference of any given point to the blue line. The more negative the residual, the more lethal the pair in the screen. The behaviour of the gRNA pairs for the nine synthetic lethal interactions common to all cell lines is shown in the A375 cell line. **D** Overlap between the synthetic lethal pairs identified in the screen in each of the three cell lines at Day 28. Significant gene pairs were defined as having a significantly more lethal pairwise effect on cellular fitness than expected from the effect of each individual gene (see Methods). Pairs where one of the genes was found to be lethal in isolation at Day 14 were excluded. **E** Percentage DNA sequence similarity between paralogues in our screen, identified as synthetic lethal in ≥1 cell line (Ever SL) vs zero cell lines (Never SL). Error bars show mean and standard deviation (SD) with each data point representing the average sequence similarity between paralogue pairs (see Source Data). The *p* value was determined using the two-tailed Mann–Whitney test.

**Benchmarking of screen performance**. Before exploring our data set for genetic interactions we elected to perform some benchmarking analyses. To do this, we first used both MAGeCK[33] and BAGEL[29] to compare the profile of gene essentiality by comparison of the screen results to established essential and non-essential genes, which were included in the library as controls (Supplementary Figs. 4 and 5). For all three cell lines, this analysis revealed a screen performance equivalent to previous large-scale single gRNA screens[34].

**Analysis of combinatorial CRISPR screen data**. The analysis to identify interacting gene pairs was based on the Bliss model of additivity[35], whereby the effect of the gRNA pair was predicted from the behaviour of each of the gRNAs by themselves (Fig. 1C). Using this approach, we calculated both the expected and observed lethality (fitness effect) of each gene pair and created a population-based model (see Methods). In this way, for each given gene pair, we were able to ascertain if the gene pair was significantly more lethal, which in this context means more depleted from the library transduced cell population, than expected by comparing the lethality of the pair to the lethality associated with individual gRNAs paired with a non-targeting/control gRNA. Using this approach, we identified between 177 and 201 candidate SL interactions per cell line, representing pairs of genes that significantly impaired cellular fitness. Given the number of SL pairs identified, we filtered the data to obtain a high-confidence set of gene pairs. First, we observed that several of the pairs identified as being potential candidate SL interactions contained a single gene with a strongly negative growth phenotype. We reasoned that this could be because some gRNAs were more efficient at disrupting their target gene or that there were subtle imbalances in the strength of the promoters used in the vector[21]. Thus, we filtered from our hits any pairs containing a gene defined as essential in either our screen (gRNA against a gene paired with a non-targeting gRNA) at day 14 (to select genes whose loss resulted in a cytotoxic rather than cytostatic effect), or an independent screen performed on each respective cell line using a whole-genome single gRNA CRISPR library[34] (see Methods, Supplementary Fig. 3). This approach refined our candidate list to 40–57 candidate SL interactions per cell line. Of the SL pairs identified in our screen ~26% (27/105) were observed in two or more cell lines and nine were found in all cell lines screened (Fig. 1D, Supplementary Data 5).

**SL is enriched between paralogous gene pairs**. Among the hits we identified, paralogues were highly over-represented (72/105 interactions, $p = 0.002$ two-tailed Fisher exact test), and all of the nine interactions common to all three cell lines were paralogues (Fig. 1D). Paralogues identified as SL gene pairs had a higher pairwise DNA sequence similarity than non-SL paralogues ($p < 0.0001$ (two-tailed Mann–Whitney test); Fig. 1E and Source Data), most likely because as homology decreases, redundancy between pair members also decreases. SL paralogues also had a higher DNA sequence similarity than non-lethal paralogues when compared with orthologues in simpler organisms (*C. elegans/D. melanogaster*) (Supplementary Fig. 6 and Source Data).

**Validation of SL pairs**. We took eight individual gene pairs identified as SL (five pairs identified as SL in 3/3 cell lines, two pairs in 2/3 cell lines, one pair in 1/3 cell line), selected based on their recurrence or biological interest, and proceeded to validate these pairs using competitive cell growth assays. To do this, we placed a gRNA targeting one of the genes in the pair in a mCherry-expressing vector, and another, targeting the other gene, in a BFP-expressing vector (see Methods). We transduced cells with both vectors to create four populations (see below), and measured population dynamics at timepoints between day 4 and 14 (Fig. 2A–B). These populations were either red or blue fluorescent, where a single gRNA had been transduced, untransduced cells with no fluorescence, or blue/red where both gRNAs had been introduced. For each cell line (A375, RPE-1, and MeWo) we established a baseline between non-interacting gene pairs by selecting two non-essential genes (*ACCSL/AIPL1*)[29] and calculated the residual, representing a baseline neutral genetic interaction in each cell line (see Methods, Supplementary Fig. 7).

We next compared the residual of each candidate SL gene pair to that of the non-interacting pair, additionally including two extra 'non-interacting pairs' as negative controls. The concordance rate between our validation experiments and the screen output was 95% (8/8 interactions in two cell lines, 7/8 in one cell line) (Fig. 2C and Source Data). Supplementary Fig. 8 and Source Data shows the fitness effects of disrupting each gene in the pair and the predicted and observed lethalities as waterfall plots.

**Comparison of the screen results to other data sets**. Across the cell lines screened in this study we identified a rich collection of SL interactions, several of which were validated as described above. To extend this analysis we compared our data to other previously generated SL data sets (Supplementary Fig. 9 and Supplementary Data 7). Specifically, we first looked for an overlap with the SL interactions computationally predicted by De Kegel et al.[8], revealing 167/234 overlapping gene pairs were in agreement with 22 of these pairs 'hits' in both studies. These pairs included *EAF1/EAF2*, *DDX39A/DDX39B* and *CHMP1A/CHMP1B*. In the same way, we compared our data set to a study by Gonatopoulos-Pournatzis et al.[22] Noting that RPE-1 was the only cell line screened in both studies, 94/170 overlapping genes pairs were concordant as either hits or non-hits. Of these, 17 were defined as SL interactions in both studies. These interactions included *CNOT7/CNOT8*, *TTC7A/TTC7B* and *SAR1A/SAR1B*, all of which are paralogous gene pairs. The gene pairs *ARID1A/ARID1B*, *SEC23A/SEC23B*, *SLC25A28/SLC25A37*, *SMARCA2/SMARCA4*, *TTC7A/TTC7B* and *UAP1/UAP1L1* were hits in all three data sets. Collectively, this analysis orthogonally validates nearly 40 SL interactions and also the quality of our screen.

**Identifying potentially therapeutically relevant genetic interactions**. We assessed the gene pairs identified as SL by our screen for their cancer-translational potential using TCGA expression data[36]. First, we searched for pairs where one member was not expressed in a tumour type reasoning that disruption of the other member of the pair may result in reduced tumour cell fitness. Interestingly, we observed *FAM50B* expression to be lost in tumours across a wide range of histological types including melanoma, bladder and colon cancer whereas it was ubiquitously expressed in normal tissue (Fig. 3A). Specifically, 355/9263 (4%) of tumours have a TPM < 1 (range 0–10% across tumour categories) (Supplementary Data 6). Of note, *FAM50B* is an imprinted gene with a paternal expression pattern, rendering it susceptible to copy number events and loss of heterozygosity[37,38]. Analysis of genome-wide methylome data revealed *FAM50B* promoter methylation in cell lines that had lost expression of the gene (Supplementary Fig. 10). Collectively, these data suggest that a proportion of human cancers could be selectively targeted by disruption or suppression of *FAM50A*.

**Fitness effect of the *FAM50A/FAM50B* genetic interaction**. We first sought to validate the *FAM50A/FAM50B* genetic interaction computationally using an independent data set[34]. Referencing whole-genome CRISPR screening data against cell line expression data suggested a significant dependency on *FAM50A* in cell lines with low or no *FAM50B* expression (TPM < 1; $p < 2.2 \times 10^{-16}$, Mann–Whitney–Wilcoxon Test) (Fig. 3B, Supplementary Data 8). We next generated isogenic knockout or isogenic "rescued" cell lines for in vitro experiments. First, using the A375 cell line, which constitutively expresses both *FAM50A* and *FAM50B*, we created a *FAM50B* knockout clone using CRISPR-Cas9 (Supplementary Fig. 11). Consequently, through use of a competitive growth assay, we found a dependency of this line on *FAM50A* (Fig. 3C and Source Data). We next took two cell lines (RKO [colorectal] and

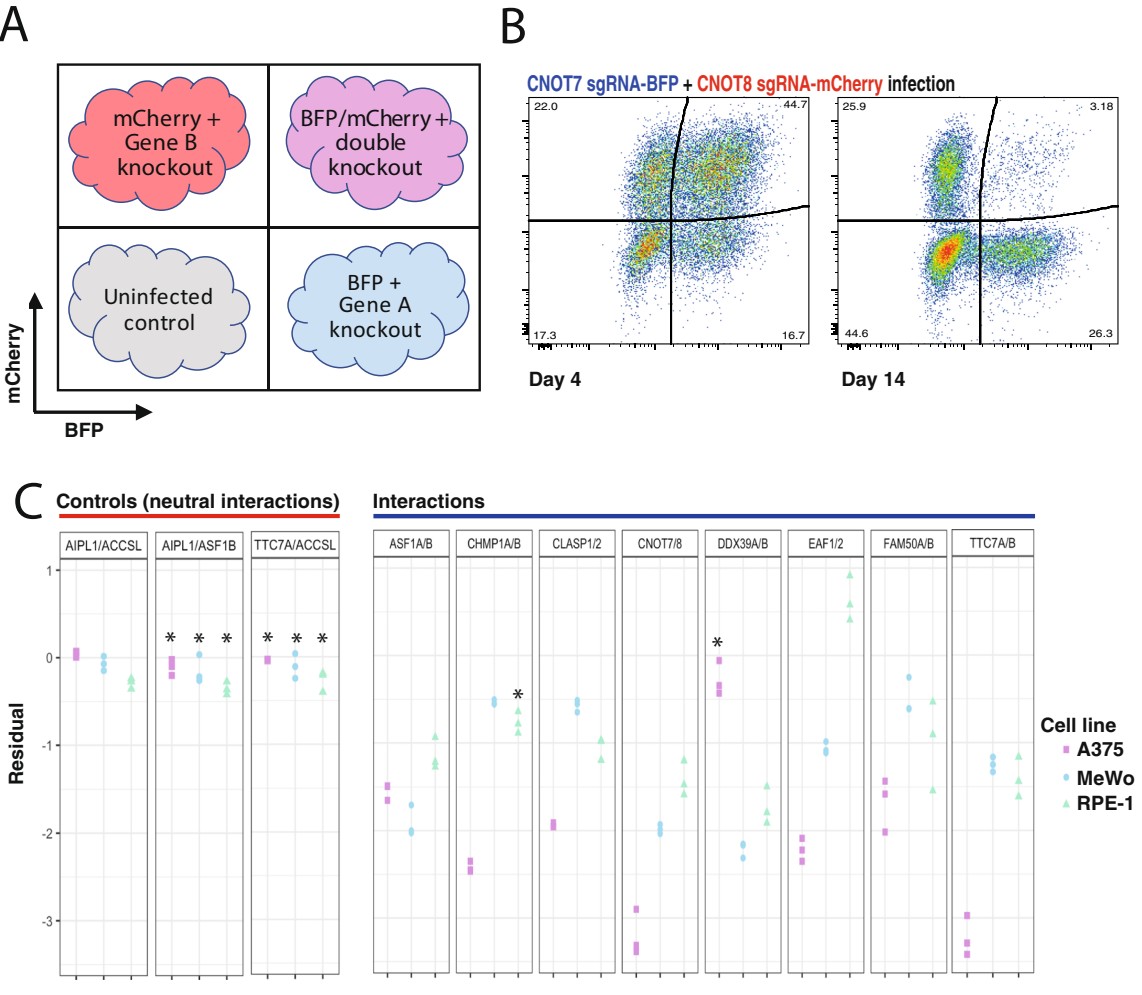

**Fig. 2 Validation of candidate synthetic lethal pairs. A** Schematic of the competitive growth assay. Cas9-expressing cells were infected with lentiviruses expressing each gRNA in a BFP or mCherry vector to create four populations and read by FACS at day 4 and 14. **B** Example of a synthetic lethal pair. A375-Cas9 cells were transduced with lentiviruses expressing *CNOT7* (BFP) and *CNOT8* (mCherry) sgRNAs and analysed at two timepoints. The growth phenotype was calculated by measuring the relative depletion of the single-infected and double-infected cells. **C** Residual values from competitive growth experiments. Residuals were calculated as the difference between the expected phenotype of the double-positive population (based on summing the phenotypes of the single positive populations) and the observed phenotype of the double-positive population. Data represent three independent transductions. The baseline residual for three non-interacting-pairs (negative controls) are also shown (*AIPL1&ACCSL, AIPL1&ASF1B, TTC7A/ACCSL*). Residuals were compared with values for the control *ACCSL/AIPL1* gene pair. Significance was calculated using a one-way ANOVA and adjusted for multiple testing using the Dunnett test. All interactions, except where indicated, had a *P* value < 0.01.

TOV21G [ovarian]) where *FAM50B* was methylated and not expressed and used lentiviral transduction to introduce a *FAM50B* cDNA construct, showing that *FAM50B* expression (F50B+) rescued the lethal phenotype associated with *FAM50A* disruption (Fig. 3C). To further assess this relationship, we performed clonogenic survival assays (Fig. 3D and Supplementary Fig. 12) in RKO cells engineered to carry a doxycycline (Dox)-inducible *FAM50A* gRNA further validating the *FAM50A/FAM50B* interaction. Thus, we have validated the genetic interaction between *FAM50A/FAM50B* using an orthogonal data set, by using isogenic *FAM50B* knockout cells, cell lines that had lost *FAM50B* expression during their evolution, and also by genetic rescue in a range of cell line models.

Of note, in our CRISPR screens we observed that inactivation of *FAM50A* alone (Supplementary Fig. 8 and Supplementary Data 1) was associated with an apparent loss of cellular fitness, despite the fact that we could readily derive *FAM50A* knockout clones (Supplementary Fig. 11). As a therapeutic target this might

suggest that inhibition of *FAM50A* could have limiting toxicity. In this regard, we recently reported a new developmental syndrome (Armfield syndrome) associated with germline hypomorphic alleles of *FAM50A* where patients developed to adulthood with phenotypes including developmental delay[39]. This observation suggests a therapeutic window for *FAM50A* inhibition. Notably, although we observed the interaction of *FAM50A/FAM50B* in all cell lines screened, many genetic interactions are highly context dependent and influenced by both the genetics of the cell line and the growth environment. Thus, further studies will be required to establish all contexts in which the *FAM50A/FAM50B* interaction is operative.

**Targeting the FAM50A/FAM50B interaction in vivo**. Not all fitness effects of gene disruption identified in culture can be replicated in vivo. To assess the possibility of targeting *FAM50A* in tumours that have lost *FAM50B* expression we used the

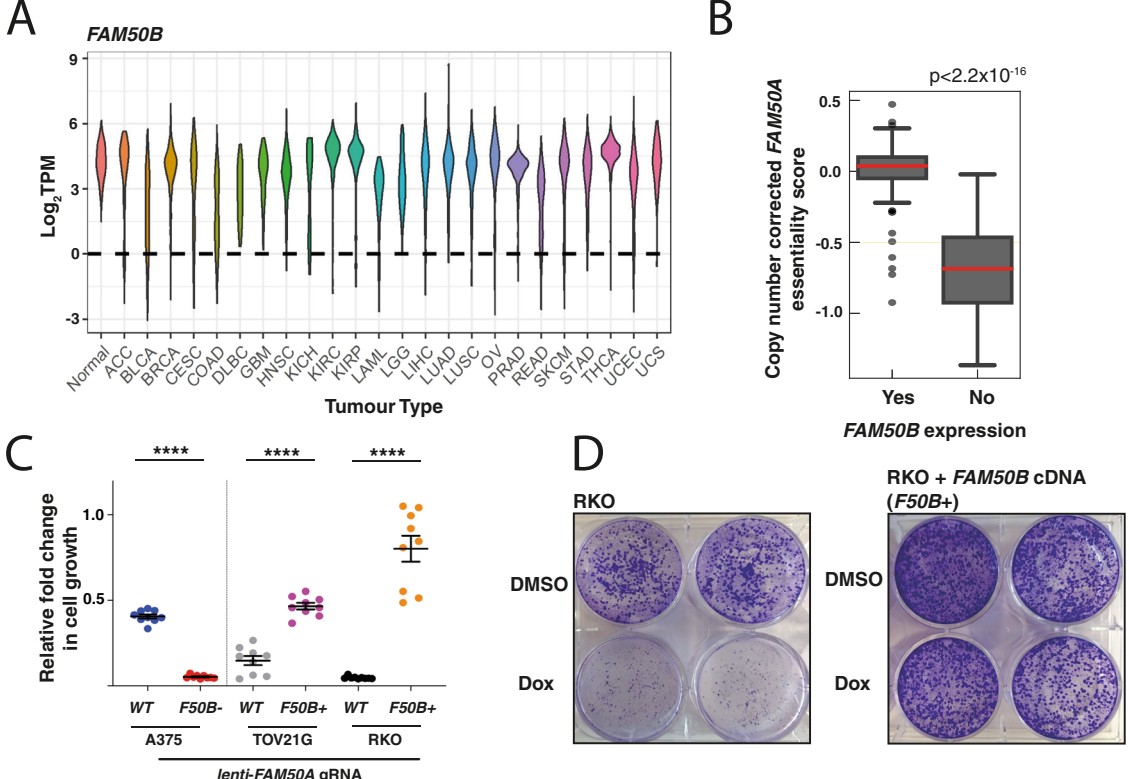

**Fig. 3 The _FAM50A/FAM50B_ axis is a targetable synthetic lethal interaction. A** Expression of _FAM50B_ across tumour and normal tissue. Data obtained from the TCGA[36]. TPM; transcripts per million. Standard TCGA tumour abbreviations where used. **B** _FAM50B_ expression inversely correlates with dependency on _FAM50A_. These data were derived from the analysis of CRISPR essentiality in 275 cancer lines[34]. Lines were categorised by their _FAM50B_ expression using an RPKM cutoff of 1 below which expression was defined as absent. The dependency of _FAM50A_ is shown as scaled $\log_2$ fold-change of _FAM50A_ gRNAs corrected for copy number variation across the cell lines used in this analysis. The _p_ value was calculated with a two-tailed Mann–Whitney _U_ test. Box-and-whisker plots show 1.5× interquartile ranges and 5–95th percentiles, centres indicate medians. **C** Analysis of essentiality using isogenic cell lines. TOV21G-Cas9 and RKO-Cas9, which do not express _FAM50B_, were transduced with a lentivirus expressing a _FAM50B_ cDNA (F50B+). A375-Cas9 cells express both _FAM50A_ and _FAM50B_. A _FAM50B_ knockout clone was generated for this cell line (F50B−) using CRISPR. Cells were transduced with a BFP+ lentivirus carrying a gRNA against _FAM50A_ such that 50% of cells were transduced. Cell proportions were analysed at day 4 and 14. The relative fold-change of the infected population is shown. _p_ values were obtained using the unpaired two-tailed _t_ test, ****$p < 0.0001$. Experiments were performed on three separate occasions in triplicate. Error bars represent the mean and the standard error of the mean (SEM). **D** Clonogenic confirmation of the _FAM50A/FAM50B_ genetic interaction. RKO and RKO-F50B+ (complemented with a _FAM50B_ cDNA) were transduced with a lentivirus containing a doxycycline-inducible gRNA against _FAM50A_ (see Methods). The cells also constitutively expressed Cas9 protein. Cells were seeded at 1000 cells/well; doxycycline (Dox; 0.1 μg/ml)/DMSO was added at 24 h. Cells were fixed and stained with crystal violet 9 days post seeding. This experiment is representative of three independent biological replicates.

TOV21G cell line carrying the Dox-inducible _FAM50A_ gRNA construct mentioned above (see Methods). As shown in Fig. 3C and 4A, disruption of _FAM50A_ in TOV21G cells precipitated a profound reduction in cellular fitness. Mice xenografted with the Dox-inducible line and fed a Dox-containing diet (0.625 g/kg; ENVIGO) exhibited a significant decrease in tumour growth compared with controls on a Dox-free diet (Fig. 4B and Source Data). We noted that after ~30 days, xenografts in which _FAM50A_ had been disrupted by administration of the Dox diet appeared to regrow (Fig. 4B). To functionally assess the mechanisms of resistance, we sequenced the transcriptomes of a collection of 34 tumours: 17 tumours from Dox-fed mice and 17 control tumours from mice fed normal chow. This analysis revealed that all tumours that regrew on Dox treatment and were collected at the ethical endpoint (1.2 cm²) showed CRISPR editing at the gRNA cut site. These edits included in-frame events, which presumably were not disruptive of _FAM50A_ but altered the cut site such that it was no longer a substrate for the gRNA, non-disruptive missense mutations and null alleles (nonsense and frameshift mutations). Wildtype traces, representing unedited

alleles, were <1% of sequence reads (Fig. 4C, D). Thus, despite efficient editing at the gRNA binding site all resistant tumours were predicted to have retained some _FAM50A_ activity. Collectively, these findings suggest that the genetic interaction between _FAM50A/FAM50B_ robustly extends to the in vivo setting.

**Loss of _FAM50A/FAM50B_ perturbs transcriptional programmes.** FAM50A and FAM50B are proteins with as-yet-undefined roles. With the exception of N-terminal coiled-coil domains they lack other recognisable sequence or structural features[40] (Supplementary Fig. 13). _FAM50A_ and _FAM50B_ have 74% DNA and 74.6% amino-acid sequence homology and are highly conserved throughout evolution. Notably, in _Chlamydomonas reinhardtii_ and _Schizosaccharomyces pombe_ orthologues of these genes have been postulated to be transcription factors or chromatin regulatory genes[40,41], whereas in human cells both FAM50A and FAM50B have been shown to interact with the C complex of the spliceosome[42,43] and via a high-throughput RNA-protein cross-linking approach, have been identified as candidate RNA-binding proteins[44].

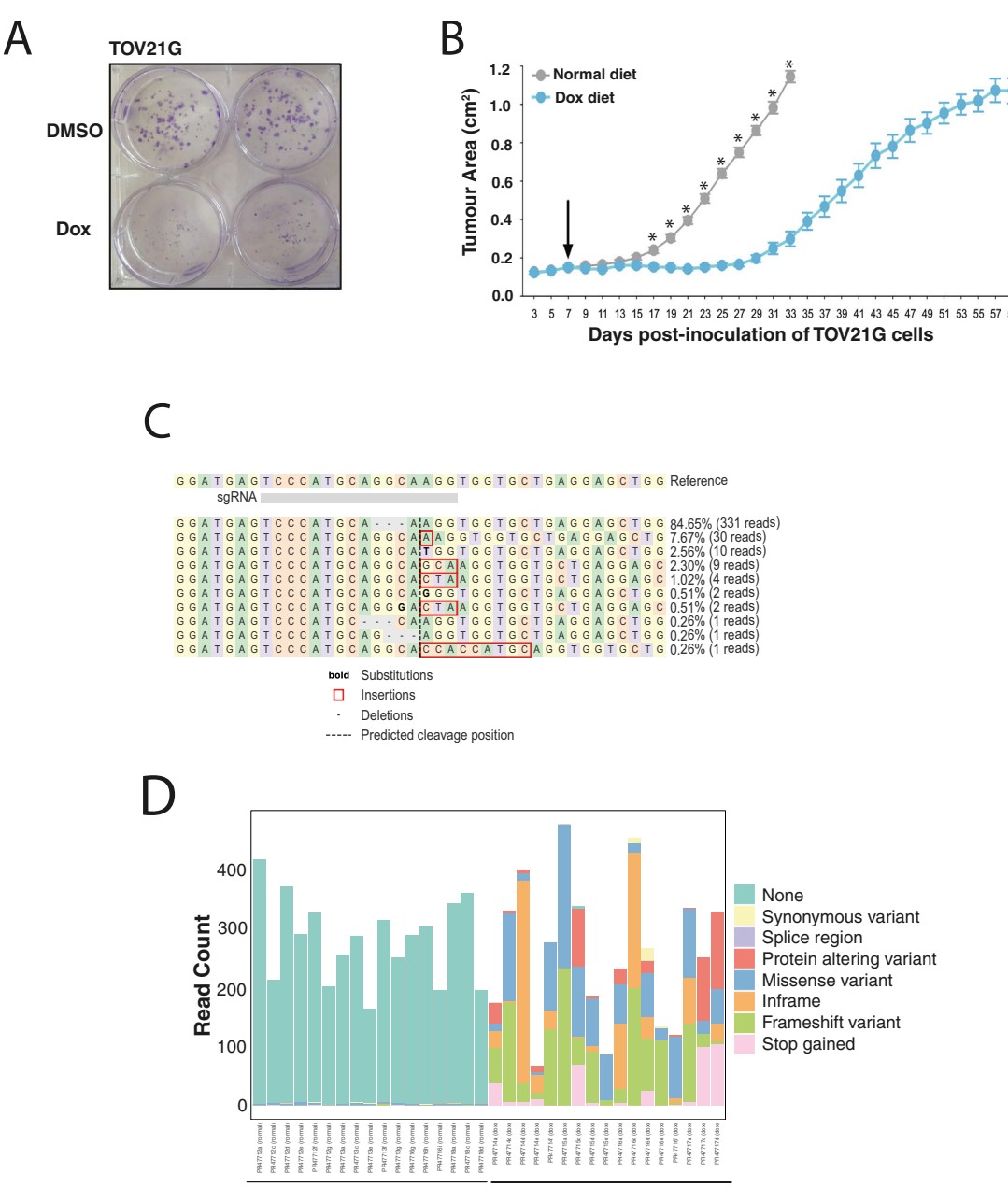

**Fig. 4 Effect of *FAM50A* loss on TOV21G cell growth in vivo and mechanisms of resistance. A** Conditional disruption of *FAM50A* in *FAM50B* non-expressing TOV21G cells engineered (see Methods) to contain a dox-inducible *FAM50A* gRNA. **B** In vivo consequence of disrupting *FAM50A* in TOV21G cells on tumour growth. Data points represent the mean and SEM (see Methods). Dox diet was administered at the day 7 timepoint (arrow). Significance was defined using a two-tailed Mann–Whitney test by comparing the area under the curve for tumour growth in the Dox-fed and control groups. *$p < 0.0001$. These data are representative of two independent experiments. **C** An example of the *FAM50A* gRNA cut site showing the profile of editing events in a doxycycline (Dox)-treated tumour after regrowth. **D** Editing outcomes at the *FAM50A* gRNA cut site in tumours collected from untreated (left) and Dox-fed mice (right). The editing events were annotated using the Variant Effect Predictor (Ensembl). The Y axis is read depth/count.

In order to assess transcriptomic changes associated with loss of *FAM50A/FAM50B*, we transcriptome profiled the effect of *FAM50A* knockout on two independent *FAM50B* null-isogenic cell line models. Using the TOV21G cell line (constitutively lacking *FAM50B* expression) or a CRISPR engineered *FAM50B* null A375 clone (A375-F50B−), we introduced a lentiviral vector carrying a gRNA to disrupt either *FAM50A* or a non-essential gene (*AIPL1*) and cultured the cells for 8 days prior to transcriptome sequencing. In both cell lines, gene set enrichment analysis revealed that disruption of *FAM50A* on a *FAM50B* null background resulted in statistically significant ($p$adj < 0.05)

transcriptional changes that included genes of the TP53 and TNFα/NFκβ pathways, and apoptosis regulators (Supplementary Fig. 14). As a comparator we also expression profiled *FAM50A* (F50A−) and *FAM50B* (F50B−) knockout A375 cells relative to parental A375 cells revealing some pathway overlaps (Supplementary Fig. 15).

**Cellular phenotypes associated with *FAM50A/FAM50B* loss.** Following an assessment of RKO cells in culture for phenotypes associated with *FAM50A/FAM50B* loss, we noted a marked increase in micronuclei, a phenotype that was rescued via the

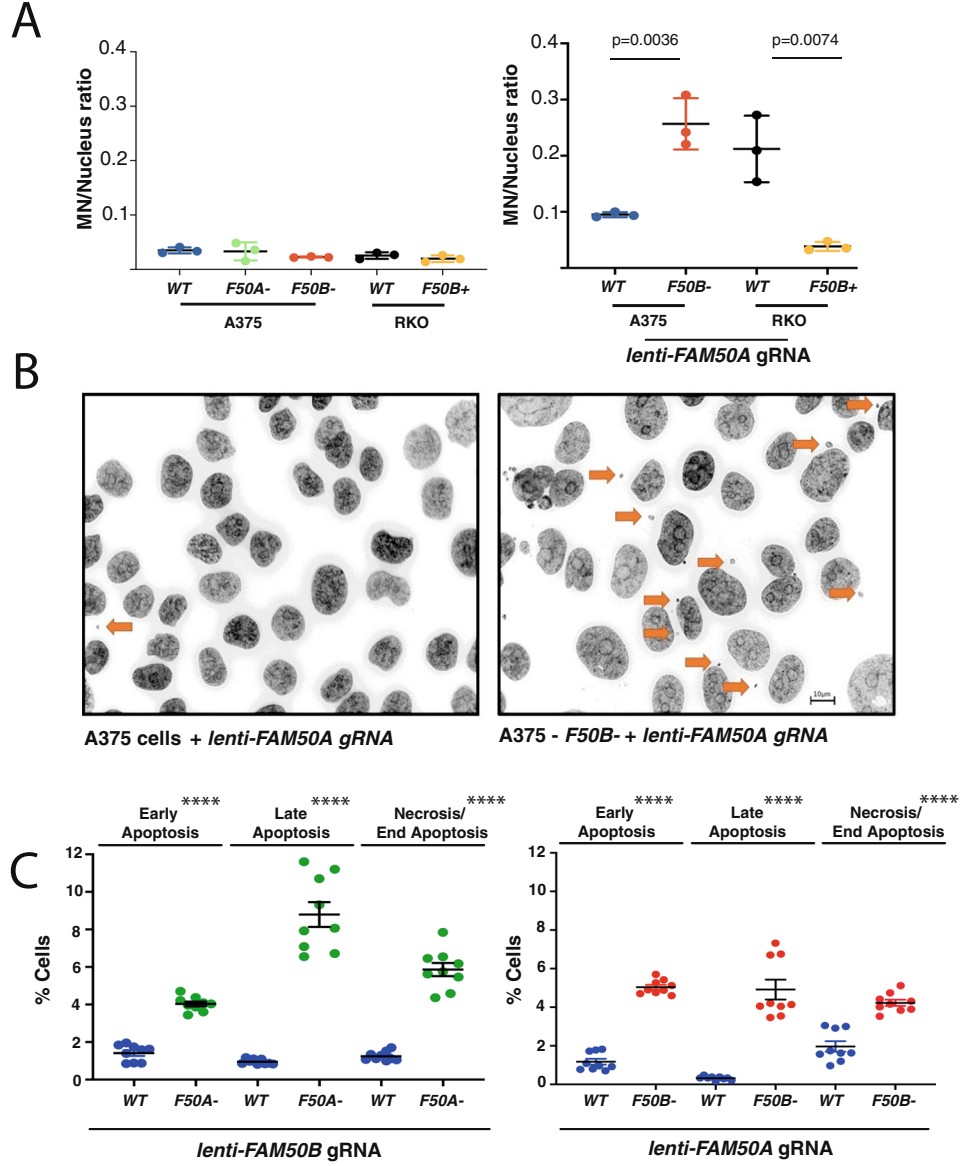

**Fig. 5 Disruption of *FAM50A/FAM50B* results in micronucleus formation and apoptosis. A** Micronucleus levels in wildtype (WT), *FAM50A* null (*F50A−*) or *FAM50B* null (*F50B−*) A375 cells and WT or *FAM50B*-complemented (*F50B+*) RKO cells (left panel). Co-disruption of *FAM50A/FAM50B* results in a profound increase in micronucleus levels that can be rescued by *FAM50B* (*F50B+*) cDNA complementation (right panel). Cells were collected for analysis 7 days after transduction. The experiment performed on three separate occasions. **B** Representative images of micronuclei in A375 cell lines from the experiment in **A**. Scale bar represents 10 μM. **C** Analysis of apoptosis at day 7–8 using the CaspGlow assay (see Methods) in A375 cells and isogenic *FAM50A* (*F50A−*) or *FAM50B* (*F50B−*) knockout derivative lines. Early apoptosis: CaspGLOW positive/viable. Late apoptosis: CaspGLOW positive/non-viable. End apoptosis or necrosis: CaspGLOW negative/non-viable. Experiments were performed on three separate occasions in triplicate. For all panels error bars represent the mean and the standard error of the mean (SEM). *p* values were calculated using the unpaired two-tailed *t* test. ****$p < 0.0001$.

introduction of the *FAM50B* cDNA (*F50B+*) (Fig. 5A, B, Source Data). We also observed enhanced micronucleus production in A375-F50B- cells transduced with a *lenti-FAM50A* gRNA compared with transduced A375 (wildtype), where loss of *FAM50A/FAM50B* also caused an induction of apoptotic cell death (Fig. 5A–C, and Source Data). Thus, loss of *FAM50A/FAM50B* causes widespread alterations in normal cellular gene expression programmes alongside micronucleation, apoptosis, and cell death.

## Discussion
Understanding the genetic wiring of cancer cells provides opportunities to identify tumour-specific vulnerabilities that might be exploited clinically. In this study, we screened >1100 gene pairs to identify 27 that were SL in multiple cell lines, several

of which we confirmed with additional validation. Intriguingly, all nine of the gene pairs identified in all cell lines screened were paralogues and paralogue pairs were significantly enriched in the collection of SL interactions we identified. This suggests that, compared with the other gene sets we analysed, which included those defined by mutual exclusivity modelling of cancer data, paralogous genes are of high value for defining synthetic lethal interactions and candidate therapeutic targets. As there are multiple agents on the market that inhibit paralogues, such as trametinib (MEK1, MEK2) and PARP inhibitors (PARP1, PARP2), each of the 27 gene pairs we identified could be considered as targets for therapy development if follow-up studies can identify selectivity for cancer cells. Another way of using essential gene pair data is to identify situations where one

member of the pair is lost somatically, thus exposing the other member of the pair as a candidate drug target and the data we have generated here could be mined for this purpose.

Of note, using TCGA expression data and referencing our collection of 27 recurrent SL gene pairs, we determined that *FAM50B* was silenced in a significant proportion of tumours across a range of histologies; 0–10% in each tumour type examined. Further, we showed that co-disruption of *FAM50A* and *FAM50B* both in vitro and in vivo precipitates a profound reduction in tumour cell fitness, a phenotype that could be partially rescued by reintroduction of *FAM50B*. Importantly, we observed dysregulation of a range of cell regulatory transcriptional programmes and the formation of extensive micronuclei together with apoptotic cell death, suggesting a specific role for *FAM50A/FAM50B* in cellular survival via maintenance of genome stability. In agreement with our study, Dede et al.[45] recently identified *FAM50A/FAM50B* as a candidate SL gene pair in cancer cell lines. Our screen is one of the first to profile gene paralogues as potential drug targets at scale and thus represents a blueprint for endeavours to prioritise all genes for screening.

## Methods

**Library construction and lentiviral production**. The CRISPR gRNA library was constructed using the method previously described by Vidigal and Ventura[31]. A detailed protocol is available at Protocols.io (dx.doi.org/10.17504/protocols.io. bpqhmmt6). The gRNAs used are provided in Supplementary Data 2 and 4. In brief, using the ENSEMBL annotation v79 of the GRCh38 version of the human reference genome we identified targeting sites in the form of 5′-NNNNNNNNN NNNNNNNNNNNNNNGG-3′, which were filtered to remove sequences that were non-unique and off-target sites using WGE[46]. Subsequently, using the ENSEMBL Perl API v84, we identified those gRNAs whose cutting sites were located within a protein domain as reported by Pfam and then discarded sequences containing BbsI restriction sites. From the resulting set of gRNAs we selected 3–5 per gene to be included in the library. Each of these gRNAs were paired with each of the 3–5 gRNAs designed for the other gene in the pair. gRNAs were also individually paired with a non-targeting Fluc_gRNA control (GTGTTGGGCGCGTTATTTATCGG) from the *Firefly* (*Photinus pyralis*) luciferase gene to allow assessment of single gene lethality. Single-targeting guides were always under the sU6 promoter (with the non-targeting gRNA under the hU6 promoter). Combinatorial guides were in a single orientation (eg hU6 guide_A + sU6 guide_B). The final library contained 41,838 different combinations of gRNAs including gRNAs targeting a total of 1191 gene pairs, 12,803 gRNAs combined with the Fluc_gRNA control and gRNAs against essential/non-essential genes to aid in statistical analysis. To make virus, $20 \times 10^6$ 293 T cells were transfected with 11.25 μg pMD2.G (Addgene #12259), 17 μg psPAX2 (Addgene #12260) and 22.5 μg of dual gRNA library plasmid. Media was changed 12 h post transduction and virus was harvested 36 h later and stored at −80 °C.

**Cell line culture and screening**. All cells were cultured at 37 °C/5%CO₂ in media as specified by ATCC. Cell line identity was confirmed by STR profiling and all cells were screened and found negative for mycoplasma. Cas9-expressing cell line generation was performed using a lentivirus produced with the pKLV2EF1a-Cas9Bsd-W plasmid (Addgene, #68343). The activity was confirmed with a BFP/GFP reporter assay[32]. All lines had ≥90% Cas9 activity prior to screening (Supplementary Fig. 2). CD15/CD33 validation experiments in Molm-13 cells (Supplementary Fig. 1) were performed using CD15-APC and CD33-PE antibodies from Miltenyi (1:5 dilution for both). The combinatorial CRISPR library was titered using cellular survival in puromycin. Library infections for screening were performed in triplicate at an MOI of 0.3, at a library representation of 1000×. Puromycin selection (2 μg/ml for A375, MeWo and 15 μg/ml for RPE-1) was continued from day 3–7 post transduction. Cells were maintained throughout the screen at a minimum representation of 3000×. Sequencing was performed on DNA extracted from these cell cultures at timepoints 14 and 28 days post transduction. As a control, wildtype A375 cells (Cas9-negative) were transduced with the library under conditions matching the screening conditions and harvested at day 7.

**Sequencing protocol**. DNA extraction was performed using a Blood & Cell Culture DNA Maxi Kit (Qiagen). For each replicate, 48× PCRs containing 3 μg genomic DNA were performed using KAPA HiFi Polymerase (ThermoFisher), using the primers listed in Supplementary Data 2. The PCRs were pooled and cleaned up with a QIAquick PCR purification kit (Qiagen) and SPRI beads (Beckman). The amplicon was then diluted to 40 pg/μl and a second round PCR was performed to add indexing primers (Illumina), using 8–10 cycles aiming for a final concentration of 4 ng/μl. The amplicon was again purified with SPRI select. The final sequencing was performed to a depth of 500-fold representation/sample,

using a customised two-forward read sequencing strategy with the primers listed in Supplementary Data 2.

**Analysis of the paired library output**. We used the Bliss independence model[35] to define synergy between gRNA pairs using the equation:

$$\text{Expected } \log_2\text{FC}(g1g2) = \text{Observed } \log_2\text{FC}(g1) + \text{Observed } \log_2\text{FC}(g2) \quad (1)$$

Note, g1g2 represents the paired gRNA construct, targeting two genes, and g1 and g2 represent each of these guides when paired with a non-targeting (Fluc) control. In this equation, the synthetic lethality/fitness effect is the difference between the log₂FC of the pair observed in the screen and log₂FC expected by the model. We performed both the control and experimental replicates in triplicate, therefore comparing each experimental replicate to each control resulted in nine comparisons per construct. For each of these nine comparisons, we plotted observed vs predicted effects, and using Loess regression, the behaviour of the population was modelled. Population modelling in this way accounts for the effect of CRISPR-mediated cutting of the genome and the potential effect of single vs double cutting. For each paired gRNA construct, a residual (vertical distance from the modelled line) was interpolated. The variance of residuals was found to be heteroscedastic, with greater variance observed when the expected effect of the guide pair was more lethal. To correct for this heteroscedasticity, variance smoothing was performed[19]. This was done by ranking each g1g2 construct by the expected fold change; the residuals were then put into bins of 200 (arbitrary number) and the variance of each bin calculated. The value of each residual was then divided by the variance of the bin that it belonged to so as to create a variance adjusted residual, resulting in equal variance across the model. For each gene pair, we then generated up to 225 (25 × 9) independent residuals from each of the gRNA paired constructs.

To derive 'hits' from the screen we required significance in two independent statistical tests; a *t* test and the robust ranking algorithm (RRA). Gene pairs typically detected in only the RRA analysis often had a single guide producing a large residual, with the remaining constructs having minimal biological effect and a mean effect close to zero. Given that in these cases, the significance of the pair was often driven by a single guide, possibly driven by off-target effects, we did not wish to consider these pairs as hits. Gene pairs typically detected in only the *t* test had a minor global negative shift, with few guide pairs ranking in the bottom 10% of residuals. Given that we were interested in pairs which had a large biological effect (favouring the RRA), but did not want to select genes where a single guide was causing outliers (favouring the *t* test) we chose to take the overlap between the two analyses to ensure size and consistency of effect.

For the *t* test the equation used was:

$$t = \frac{\text{mean (Variance adjusted residuals AB)}}{\sqrt{\frac{\text{variance (Variance adjusted residuals AB)}}{\text{n (Variance adjusted residuals AB} - 1)}}} \quad (2)$$

Robust ranking was performed as detailed previously[33] using the bottom 10% of residual values. For each test (*t* test or RRA) we used a Bonferroni test to correct for multiple testing and counted as significant pairs with an FDR < 0.1.

**Filtering candidate genetic interactions**. We noted that lethal gRNAs defined by either MAGeCK or BAGEL[29,30] analysis of the single gene targeting constructs (i.e., gRNAs paired with a non-targeting gRNA), when put under the hU6 promoter in a gene pair, were more likely to produce a significant residual ($p < 2.2 \times 10^{-6}$, Fisher exact test). This is likely due to enhanced gRNA activity in this context. In view of this, we did not consider any pairs containing genes lethal in isolation as screen hits. Genes were considered lethal if BAGEL or MAGeCK detected them as lethal in our screen at day 14, or in an independent screen[34] (using an FDR < 0.1 for MAGeCK and a PPV < 5% for BAGEL).

**Validation of candidate SL interactions**. Genetic interactions were validated using the technique established previously[19]. In total, eight pairs underwent low-throughput validation, with three additional non-interacting pairs used as controls (*AIPL1/ACCSL, AIPL1/ASF1B, TTC7A/ACCSL*). Guides showing activity across all cell lines were chosen for the validation (Supplementary Data 4) and cloned into the lentiviral pKLV2-U6gRNA5(BbsI)-PGKpuro vector backbone expressing either BFP or mCherry (Addgene #67974 or #67977). Cells were transduced in triplicate to create four populations (Fig. 2A) and the abundance of each population was read at day 4 and day 14 by FACS. Analysis was performed with FlowJo (v10.4.2) and graphs drawn with GraphPad v7.04 & v8.4.3 and R (3.6.3).

**Construction of isogenic cell lines**. To generate knockout clones, the A375-Cas9 cell line was transduced with a lentivirus containing a gRNA against *FAM50A* or *FAM50B* at single copy (Supplementary Data 2). Cells were selected with puromycin then single cells sorted by FACS into 96-well plates. Clones were expanded, and the editing site was Sanger sequenced to confirm editing. Clones chosen for further studies carried frameshift mutations (Supplementary Fig. 11). The *FAM50A* knockout clone was further confirmed by western Blot analysis (Supplementary Fig. 11) using and anti-FAM50A antibody (ABCAM, Rabbit monoclonal (ab186410) 1:3000 dilution). An anti-vinculin antibody from Sigma (SAB4200080) was used as a loading control (1:3000 dilution). An antibody for FAM50B is not

available. Both *FAM50A* and *FAM50B* clones were also validated by visual inspection of transcriptome sequence data generated as described below.

Two cell lines (RKO-Cas9 and TOV21G-Cas9) lacking *FAM50B* expression based on RNAseq data[34] were selected for rescue/complementation experiments. A lentivirus containing a full-length *FAM50B* cDNA was produced (Myc-DDK tagged; Origene; RC201531L3) and overexpression of FAM50B confirmed by western blotting (Myc tag 05-724 [clone 4A6]; Millipore, 1:2000 dilution). To create inducible knockouts, a gRNA against *FAM50A* (Supplementary Data 2) was cloned into the CRISPR pRSGTEBleo-U6Tet-(xx)-EF1-TetRep-2A-Bleo backbone (Cellecta) following the manufacturer's instructions and correct assembly confirmed by Sanger sequencing. This vector was transduced into RKO-Cas9 and TOV21G-Cas9 cells. Five days after transduction cells were single cell sorted into 96-well plates, and one week post sorting 200 μg/ml zeocin was added to the media for 3 weeks to identify resistant clones.

**Clonogenic assays.** Cells were seeded into six-well plates at 1000 cells/well. 24 h post seeding, media was changed to either contain doxycycline (0.1 μg/ml) or an equivalent volume of dimethyl sulfoxide (DMSO). The assay was terminated as cells in the control (DMSO) wells approached confluence. Cells were fixed with 100% ice-cold methanol and stained with 1% crystal violet solution.

**Apoptosis assays.** The CaspGlow assay (ThermoFisher) was used to assess apoptosis at day 7/8 after viral transduction as per the manufacturer's instructions.

**RNASeq analysis.** To assess the transcriptional effect of co-disruption of *FAM50A* and *FAM50B*, A375 *FAM50B* cells (A375-*F50B*−) and TOV21G cells were seeded at 500,000 cells per well in six-well plates and transduced with either the BFP-*F50A*_gRNA lentivirus or the BFP-*AIPL1*_gRNA lentivirus to infect >90% of cells. Media was changed at 24 h and cells were maintained in culture until harvest (day 7–8). The experiment was performed as three independent transductions per condition. RNA extraction was performed with the Direct-zol RNA microprep (Zymo research; R2060) according to manufacturers' instructions. Library preparation was performed using the KAPA stranded RNAseq kit using RiboErase. Samples were multiplexed and sequenced on a HiSeq 2500 with 75 bp paired-end reads. Sequences were pseudo-aligned to the *Homo Sapiens* transcriptome and quantified using the Kallisto quantifier[47] (v 0.44.0). We ran DESeq2[48] on the gene-level counts, first removing all genes with low-level counts. Gene set enrichment analysis was performed with fgsea (https://bioconductor.org/packages/release/bioc/html/fgsea.html).

**In vivo assessment of *FAM50A* essentiality in TOV21G xenografts.** For examination of primary tumour growth, NOD-SCID mice at 6–8 weeks of age were subcutaneously administered $2.5 \times 10^6$ TOV21G- iF50A clone 9 cells (in 0.1 mL phosphate-buffered saline; PBS) in the right flank. The mice were fed Mouse Breeders Diet (Laboratory Diets, 5021-3) and one week after dosing were randomly assigned into two cohorts with one being fed a Doxycycline diet (Dox; 625 mg/kg; Envigo, TD.01306) and the other remaining on the Mouse Breeders Diet for the remainder of the study. The developing tumours were measured every second day until they reached $1.2 \text{ cm}^2$ (calculated by: longest length measurement × longest width measurement), at which point animals were humanely sacrificed and the mass excised and stored at −80 °C. The care of mice in this study and all experimental procedures was in accordance with Home Office guidelines (P6B8058BO). Procedures were further approved by the Animal Welfare Ethical Review Body (AWERB) of the Welcome Trust Sanger Institute. Housing and husbandry conditions were exactly as detailed previously[49]. The experiment was independently replicated twice. The first cohort were female mice and the second cohort were male mice.

**Micronucleus assay.** To count micronuclei, cells were grown on coverslips, then at the time of the assay, coverslips were washed once with PBS then fixed with three parts ethanol/1 part acetic acid for one minute. Coverslips were then washed twice with PBS before mounting onto microscope slides with Vectashield DAPI mounting medium (Vector Labs). Micronuclei were counted on a Leica microscope, selecting random fields and counting all nuclei and micronuclei in each field until >500 nuclei had been counted.

All mouse experiments were performed under Home Office Project Licence P6B8058BO with ethics board approval.

**Reporting summary.** Further information on research design is available in the Nature Research Reporting Summary linked to this article.

## Data availability
The raw sequencing data are available for download from the European Genome-Phenome Archive: CRISPR data: EGAD00001006648 and RNAseq data: EGAD00001006649. All other data can be found in the Supplementary Data of this paper or in the Source Data. Source data are provided with this paper.

## Code availability
All of the code used in the paper was from published studies and is cited in the manuscript. Version numbers are provided in the Reporting Summary.

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

## Acknowledgements

N.T. was funded by the Wellcome Trust Sanger Institute Clinical PhD Programme. This work was funded by the Wellcome Trust (206194) and by grants to DJA from Cancer Research UK and the European Research Council under the European Union's Seventh Framework Programme (FP7/2007–2013)/ERC synergy grant agreement no. 319661 COMBATCANCER. Research in the SPJ lab was supported by a Cancer Research UK Programme Grant (C6/A18796), a Wellcome Investigator Award (206388/Z/17/Z), by Gurdon Institute Core Funding from Cancer Research UK (C6946/A24843), the Wellcome Trust (WT203144) and by the ERC Synergy Grant 855741 (DDREAMM). S.P.J. receives salary support from the University of Cambridge. Research in the MJG laboratory was supported by the Wellcome Trust and Open Targets. We thank Daniela Robles for help with artwork and the Sanger Institute flow cytometry core.

## Author contributions

N.A.T. and D.J.A. conceived the experiments. The majority of the experimental work was performed by N.A.T. with help from G.Z.V., M.R., A.D., A.S., F.B., J.H., E.A., B.F., L.S., S.S.M.E., F.Y., L.v.d.W., J.C., V.H., H.R., K.W., M.P.C. and M.G. S.P.J., E.G., F.I., V.I., P.C., V.O. and M.d.C.V. performed the analysis.

## Competing interests

The authors declare no competing interests.
