## [Peer Review File · Nature Communications]

Reviewers' Comments:

Reviewer #1:

Remarks to the Author:

Summary of work and major conclusions

Authors present work investigating predicted synthetic lethal interactions with focus on gene paralogues. Using a dual-guide CRISPR screening approach, the authors screen a curated selection of gene pairs across three cell lines and assess viability effects.

FAM50A/FAM50B synthetic lethality was identified and then confirmed in isogenic cell models as well as in an in vivo setting.

The authors then use gene set enrichment analysis to show that loss of FAM50A in FAM50B- cells is associated with TP53 and apoptotic pathways, as well as increased presence of micronuclei. They conclude that FAM50A/FAM50B may be a targetable dependency in cancer.

Is the question addressed important?

The broad question of identification of robust synthetic lethal interactions is important, in the context of novel therapeutic opportunities for cancer treatment.

Recent studies have suggested that paralogues may represent some of the most robust synthetic lethal interactions (e.g. ARID1A/ARID1B synthetic lethality), but that these are commonly missed in classical genetic screens (where one gene is modified per cell) due to their compensatory buffering effects.

Are the conclusions novel and will they influence thinking in the field?

Attempts to investigate the targeting of paralogues in cancer have mainly focused on known commonly mutated genes that have paralogues, such as ARID1A or SMARCA4 (with ARID1B or SMARCA2, respectively). These have largely been validated, but the question of how to specifically target individual paralogues has not been addressed (for example, in terms of drug specificity).

Although others have used combinatorial CRISPR screen approaches to identify synthetic lethal interactions, this study is, to our knowledge, the first to use the approach to systematically screen paralogues. Comparison of this data with reported synthetic lethal relationships and existing single-guide screens could be informative.

Interestingly, the authors focus on a paralogous pair of which relatively little is known (FAM50A/FAM50B) which could be of interest if mechanism is addressed.

Detailed comments on the manuscript

In general, the data is explained clearly and presented well. However, there are elements which warrant further clarification:

Major comments

The authors use a multiplex construct containing two guides with expression driven by hU6 and sU6 promoters (Figure 1B). In the methods, the authors state that guides for a given gene were all cloned to the same promoter. Is there a reason for this? Given that there is evidence that these promoters are not equally effective (something the authors themselves refer to on p5) are there any concerns that viability effects of some genes may be under- or overestimated?

The screen data presented offers a potentially useful resource to the research community. It would be informative if the authors could provide a little more QC data relating the screen to those

already published. Do individual gene effects match those previously reported in traditional CRISPR viability screens (i.e. are essential genes identified)? Can the authors offer any explanation why known synthetic lethal interactions such as ARID1A/ARID1B do not score highly in these screens?

Figure 4 is a little confusing. In 4B, is it correct that no unedited sequence was recovered in doxycycline-induced tumors, suggesting that all tumor cells were edited?

The authors begin some limited mechanistic analysis of the FAM50A/FAM50B relationship in Figure 5 in which transcriptional changes of FAM50A loss on a FAM50B-null background are assessed. Was this also performed on a FAM50B wild-type background? This may provide further insight into the gene's normal function.

Minor comments

Could the authors expand on the inclusion of the 4 miscellaneous gene pairs (Figure 1A, Supp Table 1A)?

Could residuals be plotted for gene pairs highlighted in Figure 1D for comparison?

We appreciate that plotting residuals in Figure 2C provides a concise way of plotting synthetic lethal effects. However, would it be possible to plot viability effects of individual guides alongside the combination for comparison? It would be informative to see individual viability effects to predict how individual loss might be tolerated in a therapeutic setting. Using the example of FAM50A loss in Figure 3C, A375 viability is reduced to under 50%, which may be concerning if found to be similar in somatic tissues.

Is expression of any of the other top genes lost in cancer types such as the shown for FAM50B in Figure 3A? Are any of the genes predicted to be "druggable"? This information may suggest other actionable synthetic lethal interactions that could separately be followed up.

For Figure 3D and 3E, could the authors label the cell lines to make it clear what is being induced by doxycycline?

In the methods, the authors state "an antibody for FAM50B is not available". This does not appear to be the case (for example, <https://www.abcam.com/fam50b-antibody-ab228890.html>). Do the authors mean they have been unable to get an antibody to work in this setting? This should be corrected.

Reviewer #2:

Remarks to the Author:

Summary:

The authors describe the design, construction and application of a Cas9-based dual-targeting pooled CRISPR library to explore genetic interactions (i.e. synthetic lethality) across 3 human cell lines. Gene pair sets in their library were chosen based on: (1) observation that pairs of genes had been identified as "co-lost" less frequently than expected by chance; (2) top scoring SL interactions from SynLethDB; and (3) paralogous pairs. Dual guide RNA efficiency was confirmed by FACS using gRNAs targeting CD15 and CD33. The dual-gRNA library was used to screen A375 and MeWo melanoma cell lines, as well as RPE-1 cells. Using the standard Bliss model of additivity, expected and observed lethality of each gene pair led to a population-based model, allowing the authors to ascertain whether a gene pair was significantly more lethal than expected. After some filtering, roughly ~50 candidate SL pairs were identified per cell line using this approach. Not surprisingly, paralogs were overrepresented in the hits. Validation of SL pairs indicated a

concordance rate of ~95% with the screen output.

Using TCGA data, the authors discovered that FAM50B in an imprinted gene with a paternal expression pattern, rendering it susceptible to copy-number events and LOH, and suggesting that certain human cancers could be selectively targeted by disruption or suppression of its paralog FAM50A. The authors go on to validate the SL interaction between FAM50A and FAM50B in vivo in TOV21G xenografts, and show that this double perturbation alters transcriptional programs and causes a marked increase in micronuclei.

Comments:

Overall the authors took a careful approach at designing and screening a dual gRNA Cas9 library. They also did a good job analyzing the data using expected and observed effects. This was a real strength of the paper for me. Examining synthetic lethal relationships is a rich area of biology that needs more papers like this to suss out important genetic interactions.

If understood correctly, each gene-targeting guide was paired with a non-targeting luciferase guide in order to assess the single gene perturbation effect in the CRISPR screen. However, the pairing with a non-targeting guide results in reduced DNA-damage compared to combinatorial gene-targeting constructs in which both guides induce DNA double stranded breaks. As described by numerous studies multiple DNA cuts can present a major confounding gene-independent effect as certain cell lines (such as RPE-1) are very sensitive to DNA damage (Gonatopoulos-Pournatzis et al, 2020, Brown et al, 2019, Haapaniemi, 2018). This may result in the calling of false positive genetic interactions. It is thus absolutely crucial to control for the DNA damage effect through pairing single gene-targeting guides with guides targeting intergenic regions or non-essential genes as opposed to non-targeting guides. The authors should address this in their focused validation work of individual screen hits across the three different cell lines.

Figure S1A shows unequal editing of the two guides expressed from the hU6 and sU6 promoters. Is this a guide-dependent (unequal editing efficiency) or promoter-specific effect (unequal guide expression)? To address this for their validation experiments the authors should use single- and combinatorial-targeting guides in which guides are expressed in both orientations (hU6 & sU6) to avoid capture of confounding factors due to unequal guide expression from the two promoters (see also minor comment below).

Before discussing results of combinatorial gene targeting, the authors should show quality controls such as dropout of essential and non-essential genes (both included in this library), non-targeting guides, single vs. double-targeting. Also, the performance of the single-targeting "dual guide" used in this study could be benchmarked against published data of conventional "single guide" CRISPR screen in same cell line as there are published genome-scale screens in some of the cell lines used by the authors (i.e. A375 and RPE1).

It would be interesting to compare the overlap between the paralog SL identified in this manuscript with the recently published data sets of Gonatopoulos-Pournatzis et al, 2020 and De Kegel & Ryan, 2019.

Minor comments

Abstract:

- 3rd sentence: "These analysis identified 105 gene combinations" – define what these combinations are doing (i.e. their disruption results in unpredicted loss of cellular fitness)

Introduction:

- instead of citing Refs. 2-4 it would make more sense to cite the original literature that uncovered the SL between PARP1 and BRCA1/2 (either original work or recent review of Lord & Ashworth)
- Ref. 8 describes SL between ENO1 and ENO2, not ENO3. Thus, please change citation here.
- Ref. 9 describes the targeting of DNA repair defects in BRCA deficient mutants and should not be cited in the sentence describing ENO1/3 and MEK1/2 paralogs. Rather this citation is related to the above discussed citations (PARP and BRCA1/2).
- elucidate why "cancer cell line screens are powerful tools for the identification of synthetic lethal interactions"
- when referring to "combinatorial CRISPR screening, to identify essential gene pairs" please there are number of references that were missed.
- "FAM50A/FAM50B are particularly notable amongst our collection of genetic interactions because around 4% (range 0-10%) of cancers show tumour specific loss of FAM50B expression" – unclear what the 0-10% range refers to. Loss across different tumor categories?

Results:

- Cite sources for "filtered this collection to identify genes where there was a single common orthologue in either *Caenorhabditis elegans* or *Drosophila melanogaster* and where disruption of this gene resulted in death of the organism"
- "we identified 701 gene pairs, 645 of which were amenable to targeting by CRISPR" – Does this comprise strictly two member paralogs or also paralog families with more than two members? If the latter is true, it would be interesting to discuss findings/differences between 2-member and >2 member families targeted in the screen. If library was restricted to 2-member families the selection criteria should be clearly declared.
- The BAGEL paper (Hart et al., 2016) currently cited only describes the computational approach used to analyze screens described in Hart et al., 2015. Please clarify which essential and non-essential gene set was used: RNAi-based gold standard essential (Hart et al. 2014), CRISPR-based 1,580 core essentials (Hart et al., 2015) or CRISPR-based core essential genes 2 (CEG2) (Hart et al. 2017)? Also, the reference essential gene sets are not called "BAGEL genes" but reference essential or core essential genes.
- Please clearly define orientation of library designs: are single-targeting guides screened in both orientations (under hU6 and sU6 promoters) or only under expression of sU6 promoter as indicated in Fig. 1B? What about combinatorial-targeting guides? Is each dual guide combination screened in both orientations (hU6 guide_A + sU6 guide_B AND hU6 guide_B + U6 guide_A)?
- Results of Figure 1C should also be plot on gene pair level and color coded for the 178-201 significant SL and high-confidence SL defined in the manuscript.
- Please cite source of "independent screen performed on each respective cell line using a whole genome single gRNA CRISPR library"

Combinatorial-CRISPR screening defines *FAM50A/FAM50B* as a targetable synthetic lethal gene pair:
NCOMMS-20-13240

REVIEWER COMMENTS

Reviewer #1 (Remarks to the Author):

Summary of work and major conclusions

Authors present work investigating predicted synthetic lethal interactions with focus on gene paralogues. Using a dual-guide CRISPR screening approach, the authors screen a curated selection of gene pairs across three cell lines and assess viability effects.

FAM50A/FAM50B synthetic lethality was identified and then confirmed in isogenic cell models as well as in an in vivo setting.

The authors then use gene set enrichment analysis to show that loss of *FAM50A* in *FAM50B*- cells is associated with TP53 and apoptotic pathways, as well as increased presence of micronuclei. They conclude that *FAM50A/FAM50B* may be a targetable dependency in cancer.

Is the question addressed important? The broad question of identification of robust synthetic lethal interactions is important, in the context of novel therapeutic opportunities for cancer treatment. Recent studies have suggested that paralogues may represent some of the most robust synthetic lethal interactions (e.g. *ARID1A/ARID1B* synthetic lethality), but that these are commonly missed in classical genetic screens (where one gene is modified per cell) due to their compensatory buffering effects.

Are the conclusions novel and will they influence thinking in the field?

Attempts to investigate the targeting of paralogues in cancer have mainly focused on known commonly mutated genes that have paralogues, such as *ARID1A* or *SMARCA4* (with *ARID1B* or *SMARCA2*, respectively). These have largely been validated, but the question of how to specifically target individual paralogues has not been addressed (for example, in terms of drug specificity).

Although others have used combinatorial CRISPR screen approaches to identify synthetic lethal interactions, this study is, to our knowledge, the first to use the approach to systematically screen paralogues. Comparison of this data with reported synthetic lethal relationships and existing single-guide screens could be informative.

We thank Reviewer 1 for their comments which we feel have improved the clarity of the text and the presentation of the figures in our paper. In the revised manuscript we discuss in more detail comparisons of the data we generated to other datasets. Of note, in the first version of the manuscript Figure 3B showed the association between *FAM50B* gene expression and *FAM50A* essentiality using the Project Score datasets (Behan et al., PMID: 30971826). We agree that these comparisons help strength the paper.

Interestingly, the authors focus on a paralogous pair of which relatively little is known (*FAM50A/FAM50B*) which could be of interest if mechanism is addressed.

As described in the paper we selected *FAM50A/FAM50B* for further analysis based on the observation that *FAM50B* expression is silenced in a range of tumour types which were transcriptome sequenced by the TCGA. We also selected this pair of genes because they are almost completely novel and we wanted to identify new biology. We show that loss of these genes results in extensive micronucleation and a transcriptional programme associated with enrichment of the *TP53*

Combinatorial-CRISPR screening defines *FAM50A/FAM50B* as a targetable synthetic lethal gene pair:
NCOMMS-20-13240

pathway.

Detailed comments on the manuscript

In general, the data is explained clearly and presented well. However, there are elements which warrant further clarification:

Major comments

The authors use a multiplex construct containing two guides with expression driven by hU6 and sU6 promoters (Figure 1B). In the methods, the authors state that guides for a given gene were all cloned to the same promoter. Is there a reason for this? Given that there is evidence that these promoters are not equally effective (something the authors themselves refer to on p5) are there any concerns that viability effects of some genes may be under- or overestimated?

When we constructed the library in 2016 we were unaware that the promoters were of different strengths and this was also not immediately evident from the pilot experiments we performed (Supplementary Figure. 1). Further, in the original paper that described the library generation method (Videgal et al., PMID: 25337876) this issue was not reported, nor had it been reported in the literature at the time. When we got the data back from the full screen we observed that the sU6 promoter was weaker than the hU6 promoter. Given that all of the constructs used to assess the single gene effect (gRNA paired with a non-targeting gRNA [NTC]) had the gRNA under the control of the weaker sU6 promoter and the NTC was driven by the hU6 promoter, we were concerned that we may underestimate the lethality of some genes. We therefore filtered the data as described in the methods (including removing genes defined as lethal from our previous paper; Behan et al., PMID: 30971826). This approach means that the remaining "hits" from the screen are robust but might, in some cases, mean that we have missed SL gene pairs. The orthogonal validation experiments we present in Figure 2 used gRNAs in two separate viruses under the control of the hU6 promoter and our validation rate exceeded 95% suggesting that we have adequately controlled for any viability effects resulting from different promoter strengths in our final analysis.

The screen data presented offers a potentially useful resource to the research community. It would be informative if the authors could provide a little more QC data relating the screen to those already published. Do individual gene effects match those previously reported in traditional CRISPR viability screens (i.e. are essential genes identified)? Can the authors offer any explanation why known synthetic lethal interactions such as *ARID1A/ARID1B* do not score highly in these screens?

We have included more QC data in the revised manuscript (Supplementary Figures 4/5). (including ROC curves for comparisons with essential genes and Bayes factor graphs). Notably, disruption of *ARID1A/1B* was found to be synthetic lethal in the Mewo cell line (Supplementary Table 5). In the A375 cell line, *ARID1A* KO alone was found to have a strong negative growth phenotype and this gene pair was therefore discounted for the reasons outlined above. The consequences of disrupting *ARID1A/1B* in RPE1 cells was recently assessed in another study (Gonatopoulos-Pournatzis et al., PMID: 32249828) and just as we report it was not found to be a synthetic lethal interaction. Collectively, these observations support the view that many synthetic lethal interactions are highly context dependent as we discuss in our manuscript.

Figure 4 is a little confusing. In 4B, is it correct that no unedited sequence was recovered in doxycycline-induced tumors, suggesting that all tumor cells were edited?

Combinatorial-CRISPR screening defines *FAM50A/FAM50B* as a targetable synthetic lethal gene pair: NCOMMS-20-13240

We thank reviewer 1 for this comment. Greater than 99% of sequences were edited suggesting, as deduced by the reviewer, that almost all tumour cells were edited. We have added a sentence to clarify this point in the revised manuscript.

The authors begin some limited mechanistic analysis of the *FAM50A/FAM50B* relationship in Figure 5 in which transcriptional changes of *FAM50A* loss on a *FAM50B*-null background are assessed. Was this also performed on a *FAM50B* wild-type background? This may provide further insight into the gene's normal function.

We think the reviewer is referring to the data in Supplementary Figure 14 which shows the gene pathways perturbed when a *FAM50B* null cell line has *FAM50A* knocked out. In the revised manuscript we also include an analysis we performed where we define the transcriptional changes associated with loss of *FAM50A* or *FAM50B* when compared to wildtype A375 cells. These data are available in Supplementary Figure 15.

Minor comments

Could the authors expand on the inclusion of the 4 miscellaneous gene pairs (Figure 1A, Supp Table 1A)?

Dr. Phil Chapman, a co-author in the paper, has been developing PARG inhibitors for prostate cancer and when we made the library he had preliminary evidence of a genetic interaction between *PARG* and *XRCC1*, so we thought to include this pair. Using a similar hypothesis driven rationale, we also chose to include gRNAs against *KDM6A/KDM6B*; paralogs involved in histone demethylation. *KDM6A* is commonly deleted or inactivated in a range of cancers (PMID: 25719666). Interestingly, this pair was not identified by our pipeline due to the presence of a third paralog member, *UTY*, situated on the Y chromosome. We subsequently chose to include the whole paralog family in our screen (*KDM6A/UTY*; *KDM6A/KDM6B*; *KDM6B/UTY*). We have clarified this point in the methods.

Could residuals be plotted for gene pairs highlighted in Figure 1D for comparison?

As requested we have replotted this figure and highlighted the significant gene pairs as requested. We decided to show the 9 interactions (225 gRNA combinations) found in all cell lines because if we plot all interactions the figure becomes unintelligible.

We appreciate that plotting residuals in Figure 2C provides a concise way of plotting synthetic lethal effects. However, would it be possible to plot viability effects of individual guides alongside the combination for comparison? It would be informative to see individual viability effects to predict how individual loss might be tolerated in a therapeutic setting. Using the example of *FAM50A* loss in Figure 3C, A375 viability is reduced to under 50%, which may be concerning if found to be similar in somatic tissues.

Reviewer 1 makes a very good point and we have provided these data in Supplementary Figure 8. Of note, we have recently been involved in a study of a group of patients with Armfield Syndrome who carry hypomorphic alleles of *FAM50A* (paper accepted in Nature Comms; NCOMMS-19-16909). These patients show developmental delay and learning difficulties but are otherwise medically well and developed to adulthood. Thus, it would be anticipated, based on this genetic evidence, that an inhibitor of *FAM50A* would be well tolerated. We have discussed this point in the revised manuscript.

Is expression of any of the other top genes lost in cancer types such as the shown for *FAM50B* in Figure 3A?

Combinatorial-CRISPR screening defines *FAM50A/FAM50B* as a targetable synthetic lethal gene pair:
NCOMMS-20-13240

We elected to work on *FAM50A/FAM50B* because *FAM50B* expression was silenced across a range of cancers. We did not see direct evidence for a similar expression pattern for the other genes i.e. loss of expression of one of the genes in the pair. However, these gene pairs might be of therapeutic interest if they are essential for the survival of a particular cellular lineage, a question that will need to be answered by screening additional cell lines in the future.

Are any of the genes predicted to be “druggable”? This information may suggest other actionable synthetic lethal interactions that could separately be followed up.

As we mention in the paper, *FAM50A* and *FAM50B* proteins have coiled-coil domains and through these sequences are thought to participate in protein-protein interactions. This pair of genes is therefore potentially druggable. We have an ongoing collaboration with Astra Zeneca which will explore the druggability of some of these genes and as a result of this question we have expanded the discussion of these genes and their potential tractability as therapeutic targets.

For Figure 3D and 3E, could the authors label the cell lines to make it clear what is being induced by doxycycline?

We thank Reviewer 1 for this comment. We have amended the manuscript accordingly.

In the methods, the authors state “an antibody for *FAM50B* is not available”. This does not appear to be the case (for example, <https://www.abcam.com/fam50b-antibody-ab228890.html> [abcam.com]). Do the authors mean they have been unable to get an antibody to work in this setting? This should be corrected.

This is an excellent point – we tried the Abcam antibody but we could not get it to work. We have clarified this point in the revised manuscript. The Western blot we performed using this antibody is shown on the Abcam website (i.e. as submitted by the 1st author). Of note, in the revised manuscript we have made clear that we visually inspected transcriptome sequence data generated from knockout clones which confirmed insertion of the editing events and also that the lines were free from wildtype contaminating cells.

We thank reviewer 1 for their very helpful comments.

Reviewer #2 (Remarks to the Author):

Summary:

The authors describe the design, construction and application of a Cas9-based dual-targeting pooled CRISPR library to explore genetic interactions (i.e. synthetic lethality) across 3 human cell lines. Gene pair sets in their library were chosen based on: (1) observation that pairs of genes had been identified as “co-lost” less frequently than expected by chance; (2) top scoring SL interactions from SynLethDB; and (3) paralogous pairs. Dual guide RNA efficiency was confirmed by FACS using gRNAs targeting CD15 and CD33. The dual-gRNA library was used to screen A375 and MeWo melanoma cell

Combinatorial-CRISPR screening defines *FAM50A/FAM50B* as a targetable synthetic lethal gene pair: NCOMMS-20-13240

lines, as well as RPE-1 cells. Using the standard Bliss model of additivity, expected and observed lethality of each gene pair led to a population-based model, allowing the authors to ascertain whether a gene pair was significantly more lethal than expected. After some filtering, roughly ~50 candidate SL pairs were identified per cell line using this approach. Not surprisingly, paralogs were overrepresented in the hits. Validation of SL pairs indicated a concordance rate of ~95% with the screen output.

Using TCGA data, the authors discovered that *FAM50B* is an imprinted gene with a paternal expression pattern, rendering it susceptible to copy-number events and LOH, and suggesting that certain human cancers could be selectively targeted by disruption or suppression of its paralog *FAM50A*. The authors go on to validate the SL interaction between *FAM50A* and *FAM50B* in vivo in TOV21G xenografts, and show that this double perturbation alters transcriptional programs and causes a marked increase in micronuclei.

Comments:

Overall the authors took a careful approach at designing and screening a dual gRNA Cas9 library. They also did a good job analyzing the data using expected and observed effects. This was a real strength of the paper for me. Examining synthetic lethal relationships is a rich area of biology that needs more papers like this to suss out important genetic interactions.

We thank Reviewer 2 for their very helpful comments and for recognising the enormous amount of careful work that was involved in this paper.

If understood correctly, each gene-targeting guide was paired with a non-targeting luciferase guide in order to assess the single gene perturbation effect in the CRISPR screen. However, the pairing with a non-targeting guide results in reduced DNA-damage compared to combinatorial gene-targeting constructs in which both guides induce DNA double stranded breaks. As described by numerous studies multiple DNA cuts can present a major confounding gene-independent effect as certain cell lines (such as RPE-1) are very sensitive to DNA damage (Gonatopoulos-Pournatzis et al, 2020, Brown et al, 2019, Haapaniemi, 2018). This may result in the calling of false positive genetic interactions. It is thus absolutely crucial to control for the DNA damage effect through pairing single gene-targeting guides with guides targeting intergenic regions or non-essential genes as opposed to non-targeting guides. The authors should address this in their focused validation work of individual screen hits across the three different cell lines.

The model we used to analyse the output of the paired screen (Bliss independence with LOESS regression model) did not make any *a priori* assumptions about the expected effect of inducing two editing events. Instead, the expected effect was derived from the behaviour of the gRNA population as a whole in each cell line independently. If, therefore, a particular cell line was particularly sensitive to double strand breaks, then all the double editing events would have a more negative log fold change (LFC), which would then shift the fitted curve of expected behaviour downwards thus adjusting for this. i.e. Our analysis approach looks for gRNA pairs which are outliers from the population as a whole and accounts for the possible effects of double cutting. We have made this point clear in the revised manuscript.

To further mitigate for the risk of “false positive” SL interactions the validation experiments shown in Figures 2 & 3 used controls that cut the genome (i.e. safe-targeting controls) and the experiments in Figures 4 & 5 used isogenic cell lines which allowed us to treat all of the cells identically thus mitigating any risk of exacerbated lethality as a result of “double cutting”.

Combinatorial-CRISPR screening defines *FAM50A/FAM50B* as a targetable synthetic lethal gene pair: NCOMMS-20-13240

Figure S1A shows unequal editing of the two guides expressed from the hU6 and sU6 promoters. Is this a guide-dependent (unequal editing efficiency) or promoter-specific effect (unequal guide expression)? To address this for their validation experiments the authors should use single- and combinatorial-targeting guides in which guides are expressed in both orientations (hU6 & sU6) to avoid caption of confounding factors due to unequal guide expression from the two promoters (see also minor comment below).

As discussed in the methods of the manuscript we determined that the unequal editing is probably promoter dependent and we rigorously account for this in our analysis pipeline.

Notably, in the validation experiments (Figure 2) we did not use paired gRNA constructs instead electing to use two independent single gRNA constructs both driven by the hU6 promoter but carrying different fluorophores (to account for “double cutting” these experiments used non-targeting controls [gRNAs that cut an inert locus]). Therefore, the potential confounding effect of promoter imbalances was eliminated in these focused validation experiments (also in the experiments in Figures 3,4 and 5). This was also a motivation for generating isogenic cell lines to further characterise the *FAM50A/FAM50B* interaction, which allowed us to deliver a single gRNAs into cells in culture.

Before discussing results of combinatorial gene targeting, the authors should show quality controls such as dropout of essential and non-essential genes (both included in this library), non-targeting guides, single vs. double-targeting. Also, the performance of the single-targeting “dual guide” used in this study could be benchmarked against published data of conventional “single guide” CRISPR screen in same cell line as there are published genome-scale screens in some of the cell lines used by the authors (i.e. A375 and RPE1).

This is a very good point – to address this question we provide QC metrics in Supplementary Figures 4 & 5 in the revised manuscript. Collectively, these analyses confirm that our dual CRISPR screens were of high quality. As detailed below our comparisons to overlapping gene pairs in other combinatorial screens confirms the quality of our screen.

It would be interesting to compare the overlap between the paralog SL identified in this manuscript with the recently published data sets of Gonatopoulos-Pournatzis et al, 2020 and De Kegel & Ryan, 2019.

This is an excellent suggestion.

Across the cell lines screened in this study we identified a rich collection of synthetic lethal interactions. To extend this analysis we compared our collection of candidate synthetic lethal interactions to other previously generated datasets (Supplementary Fig. 9 & Supplementary Table 7). Specifically, we looked for an overlap with the 105 candidate synthetic lethal interactions we identified and interactions computationally predicted by De Kegel et al.,⁸. This analysis revealed that 167/234 overlapping gene pairs agreed in both studies (22 common hits). These pairs included *EAF1/EAF2*, *DDX39A/DDX39B* and *CHMP1A/CHMP1B*. In the same way we compared synthetic lethal interactions identified in our study and a paper by Gonatopoulos-Pournatzis *et al.*,²³. Noting that RPE-1 was the only cell line screened in both studies, 94/170 genetic interactions agreed (17 common hits). These interactions included *CNOT7/CNOT8*, *TTC7A/TTC7B* and *SAR1A/SAR1B*, all of which are paralogous gene pairs. Collectively, this analysis orthogonally validates nearly 40 synthetic lethal interactions and also the quality of our screen.

Minor comments

Abstract:

- 3rd sentence: “These analysis identified 105 gene combinations” – define what these combinations are doing (i.e. their disruption results in unpredicted loss of cellular fitness)

This is a good point and we have amended the manuscript accordingly.

Introduction:

- instead of citing Refs. 2-4 it would make more sense to cite the original literature that uncovered the SL between PARP1 and BRCA1/2 (either original work or recent review of Lord & Ashworth)

We have done this as suggested. We agree it’s best to cite the original references.

- Ref. 8 describes SL between ENO1 and ENO2, not ENO3. Thus, please change citation here.

We have edited this citation as suggested. Thank you for pointing this out.

- Ref. 9 describes the targeting of DNA repair defects in BRCA deficient mutants and should not be cited in the sentence describing ENO1/3 and MEK1/2 paralogs. Rather this citation is related to the above discussed citations (PARP and BRCA1/2).

We have edited this citation accordingly.

- elucidate why “cancer cell line screens are powerful tools for the identification of synthetic lethal interactions”

We have expanded on this point in the revised manuscript to further clarify why cancer cell lines screens are a useful discovery tool.

- when referring to “combinatorial CRISPR screening, to identify essential gene pairs” please there are number of references that were missed.

We have done our best to include all of the papers in the revised manuscript including those published very recently.

- “*FAM50A/FAM50B* are particularly notable amongst our collection of genetic interactions because around 4% (range 0-10%) of cancers show tumour specific loss of *FAM50B* expression” – unclear what the 0-10% range refers to. Loss across different tumor categories?

We have edited the text between the brackets so that is reads as follow “(range 0-10% dependent on tumour type)” to indicate that the 0-10% indicates the frequency of tumours that have lost *FAM50B* expression.

Results:

- Cite sources for “filtered this collection to identify genes where there was a single common

Combinatorial-CRISPR screening defines *FAM50A/FAM50B* as a targetable synthetic lethal gene pair: NCOMMS-20-13240

orthologue in either *Caenorhabditis elegans* or *Drosophila melanogaster* and where disruption of this gene resulted in death of the organism”

The Version of Flymine accessed was FB2015_05; the version of Wormbase accessed was WS251. We have added these details to the revised manuscript.

- “we identified 701 gene pairs, 645 of which were amenable to targeting by CRISPR” – Does this comprise strictly two member paralogs or also paralog families with more than two members? If the latter is true, it would be interesting to discuss findings/differences between 2-member and >2 member families targeted in the screen. If library was restricted to 2-member families the selection criteria should be clearly declared.

We restricted the paralog gene pairs to those where there were only two genes. We have made this point clear in the revised manuscript.

- The BAGEL paper (Hart et al., 2016) currently cited only describes the computational approach used to analyze screens described in Hart et al., 2015. Please clarify which essential and non-essential gene set was used: RNAi-based gold standard essential (Hart et al. 2014), CRISPR-based 1,580 core essentials (Hart et al., 2015) or CRISPR-based core essential genes 2 (CEG2) (Hart et al. 2017)? Also, the reference essential gene sets are not called “BAGEL genes” but reference essential or core essential genes.

We connected with Traver Hart and he told us to cite the Hart 2017, G3, that describes the CEGv2 reference essentials. We have also amended the text so we don’t refer to the genes as BAGEL genes rather essential and non-essential genes as suggested. We also provide a link to the exact gene lists we used.

- Please clearly define orientation of library designs: are single-targeting guides screened in both orientations (under hU6 and sU6 promoters) or only under expression of sU6 promoter as indicated in Fig. 1B? What about combinatorial-targeting guides? Is each dual guide combination screened in both orientations (hU6 guide_A + sU6 guide_B AND hU6 guide_B + U6 guide_A)?

The single gRNAs targeting genes were under the control of the sU6 promoter and paired with a non-targeting control driven by a hU6 promoter. The dual guides are in both orientations – we have clarified this point in the methods of the revised manuscript.

- Results of Figure 1C should also be plot on gene pair level and color coded for the 178-201 significant SL and high-confidence SL defined in the manuscript.

As requested we have replotted this figure and highlighted the significant gene pairs. We decided to show the 9 interactions (225 gRNA combinations) found in all cell lines because if we plot all interactions the figure becomes unintelligible.

- Please cite source of “independent screen performed on each respective cell line using a whole genome single gRNA CRISPR library”

This was Behan et al., (PMID: 30971826). We have made this clear in the revised manuscript.

We than Reviewer 2 for their very helpful comments.

Combinatorial-CRISPR screening defines *FAM50A*/*FAM50B* as a targetable synthetic lethal gene pair:
NCOMMS-20-13240

Reviewers' Comments:

Reviewer #1:

Remarks to the Author:

The authors have addressed each fo the points I raised in a very satisfactory way. I'd like to thank the authors for providing a concise response as well. I think this will make a great addition to the literature.

Reviewer #3:

Remarks to the Author:

The authors have sufficienctly addressed the comments

REVIEWERS' COMMENTS

Reviewer #1 (Remarks to the Author):

The authors have addressed each fo the points I raised in a very satisfactory way. I'd like to thank the authors for providing a concise response as well. I think this will make a great addition to the literature.

We thank Reviewer 1 for their advice during the review process and thank them for their comments.

Reviewer #3 (Remarks to the Author):

The authors have sufficienctly addressed the comments

We thank Reviewer 3 for engaging in the review process and for their comments on our manuscript.